# Barriers and motivators of contraceptive use among young people in Sub-Saharan Africa: A systematic review of qualitative studies

**Luchuo Engelbert Bain** [1,2]*, **Hubert Amu** [3], **Elvis Enowbeyang Tarkang** [3]

**1** Lincoln International Institute for Rural Health (LIIRH), College of Social Science, University of Lincoln, Lincoln, Lincolnshire, United Kingdom, **2** Global South Health Research and Services, GSHS, Amsterdam, The Netherlands, **3** Department of Population and Behavioural Sciences, School of Public Health, University of Health and Allied Sciences, Hohoe, Ghana

* lebaiins@gmail.com

**Data Availability Statement:** All relevant data are in the paper and its Supporting Information files.

**Funding:** The authors received no specific funding for this work.

## Abstract

### Background

In sub-Saharan Africa, about 80% of young women either use a traditional method or do not use any form of contraception at all. The objectives of this review were to ascertain the barriers and motivators of contraceptive use among young people in Sub–Saharan Africa.

### Materials and methods

We conducted electronic literature searches in PubMed, EMBASE, Ebsco/PsycINFO and Scopus. We identified a total of 4,457 publications and initially screened 2626 based on the Preferred Reporting Items for Systematic Reviews and Meta-Analyses (PRISMA). A total of 13 qualitative studies were retained for the final analysis based on the Joanna Briggs criteria for assessing qualitative studies. The systematic review is registered on PROSPERO with identifier CRD42018081877.

### Results

Supportive social networks, respect for privacy and confidentiality, ready availability, affordability and accessibility of contraceptives, as well as the desire to prevent unintended pregnancy and sexually transmitted infections were the motivators of contraceptive use among young people in sub-Saharan Africa. Despite these motivators, myriad of personal, societal, and health systems-based barriers including myths and misconceptions, known side effects of contraceptives, prohibitive social norms, and negative attitude of health professionals were the major barriers to contraceptive use among young people.

### Conclusion

Sub-Saharan African countries with widespread barriers to contraceptive use among young people may not be able to achieve the Sustainable Development Goal 3.8 target of achieving health for all by the year 2030. Interventions intended to improve contraceptive use need

**Competing interests:** The authors have declared that no competing interests exist.

**Abbreviations:** FGD, Focus Group Discussion; LARC, Long-Acting Reversible Contraceptives; PRISMA, Preferred Reporting Items for Systematic Reviews and Meta-Analyses; STI, Sexually Transmitted Infection; SDG, Sustainable Development Goal; SSA, Sub-Saharan Africa; WHO, World Health Organisation.

to be intersectoral and multi-layered, and designed to carefully integrate the personal, cultural, organizational and political dimensions of contraception.

## Introduction

Pregnancy and childbirth among young women remain major public health concerns worldwide [1]. In low- and middle-income countries (LMICs), about 16 million girls aged 15 to 19 years give birth every year, with about 2.5 million being under the age of 16 years [1]. In sub-Saharan Africa (SSA), about 13% of pregnancies end up in abortions and 97% of these are unsafe [2]. Optimal contraceptive use alone has the capacity of reducing the burden of unintended pregnancies and abortions by one third [3, 4]. High levels of contraceptive failure or discontinuation are accrued predominantly to the use of traditional methods (coitus interruptus [withdrawal method], lactational amenorrhea method, calendar/rhythm method, cervical mucus method, and abstinence), and consequently increase the overall burden of unintended pregnancies and abortions in SSA [4].

With many SSA countries still having challenges the provision of optimal access to safe abortion care, non–use or inconsistent use of contraceptives account for most of the unsafe abortions. The effects are high maternal mortality and morbidity rates, especially among young people [1]. The contraceptive challenges and their resultant morbidities and mortalities occur at the backdrop that SSA countries have signed up to meeting the Sustainable Development Goal (SDG) 3.8 target of achieving health for all by the year 2030 [5, 6]. Achieving health for all, therefore, implies that the countries have to, among other things increase the level of contraceptive use among their populace especially young people. Young people according to the World Health Organisation [7] are people 10–24 years old. Intrauterine devices and contraceptive implants, also called Long-Acting Reversible Contraceptives (LARC) are the most effective reversible contraceptive methods [8] and are highly recommended for young people [9, 10].

Despite high levels of awareness that have been reported in the literature regarding contraceptives in SSA, utilisation has been overwhelmingly sup-optimal [11–16]. In SSA, however, about 80% of young people use traditional methods or do not use any form of contraception at all [8]. This review was, therefore, guided by two objectives which were to ascertain the barriers to contraceptive use and explore the motivators of contraceptive use among young people in SSA. Findings from this review could be potentially relevant in adequately planning and implementing public health interventions among this target group.

## Materials and methods

A comprehensive search was performed in the bibliographic databases: PubMed, Embase.com, Ebsco/PsycINFO and Scopus in collaboration with the medical librarian of the Vrije Universiteit, Amsterdam. A review protocol was developed based on the Preferred Reporting Items for Systematic Reviews and Meta-Analyses (PRISMA)-statement (www.prisma-statement.org). Databases were searched from inception up to 10th September 2019. The following terms were used (including synonyms and closely related words) as index terms or free-text words: "Africa South of the Sahara", "sub-Saharan Africa" "Adolescent", "Young women", "young people" "Family Planning", "Contraception", (contraceptives) "Birth Control", "young female", "teen", "family planning", "barrier", "factor", "enablers", "facilitators", "predictor", "determinant". The search was performed without date, language or publication status restriction. Duplicate

articles were excluded. The full search strategies for all databases can be found in the Supplementary Information. The systematic review is registered on PROSPERO with identifier CRD42018081877.

Only qualitative studies were retained for the final review. Our decision to include only qualitative studies was informed by the paucity of qualitative analyses especially in terms of reviews in SSA. We thus sought to bridge this gap by providing a deeper and more holistic appreciation of the barriers and motivators of young people's contraceptive use. To ensure the trustworthiness of the themes and sub-themes generated, two independent researchers were involved in the selection of articles. The reviewers compared and decided upon the articles to be retained for final review after discussing and coming to a consensus. In addition to these, two reviewers assessed clear reporting of aims and objectives of the study, adequate description of the context in which the research was carried out, adequate description of the sample and the methods by which the sample was identified and recruited, adequate description of the methods used to collect data, and adequate description of the methods used to analyse data. A third reviewer was on standby to be involved if disagreement happens between the two reviewers. The Joanna Briggs criteria for assessing qualitative studies were used [17] with a particular focus on the critical appraisal checklist for qualitative research, to determine the eligibility of studies included in the final review. Methodological insights reported by Uman (2011) were applied in the data interpretation and discussion phases of the study [18]. We used conceptual thematic analysis to re-categorise the barriers and motivators reported by the studies into major themes, sub-themes, and their respective codes. These were then organised in the form of tables.

## Results

### Background of studies included in the review

Electronic search through relevant databases yielded a total of 4,457 publications. Out of this, 2,626 articles were screened based on article title and abstracts. At this stage, 2,475 were excluded due to unfit title and abstract. The remaining 151 full-text articles were further screened for eligibility and 134 were expunged. A total of 13 studies were finally retained. Fig 1 presents the PRISMA flow diagram of the literature selection. The studies spanned the period 2000 to 2018 (See Table 1). Four were conducted in Ghana and two in South Africa. Mali, Nigeria, Uganda, Kenya, Tanzania, Mozambique, and Senegal each recorded one study. While 12 of the studies generally targeted young people, one focused on young people with disabilities. Concerning methods of data collection, seven of the studies used only in-depth interviews, three studies used only focus group discussions, and one adopted IDI and FGD. Of the remaining two studies, one added responses of young people in a general discussion to FGDs while the other one added informal conversation and observation to IDI and FGD. For 12 of the studies, sample sizes used ranged from 15 to 149. Ten of the studies focused on all contraceptives while one focused on condoms. The remaining two also focused on emergency contraceptives and modern contraceptives respectively (See Table 1).

### Motivators of contraceptive use

Eight out of the 13 studies reported on motivators of contraceptive use. Table 2 presents the motivators of contraceptive use as reported by the eight studies. Five main themes were realised from the review. These were: social support, protection of identity, ready availability, affordability and access, effectiveness in preventing unintended pregnancy and Sexually Transmitted Infections (STIs), and other motivations. The "other" theme was labelled as such because the motivations under the theme were mentioned by just one study in each case.

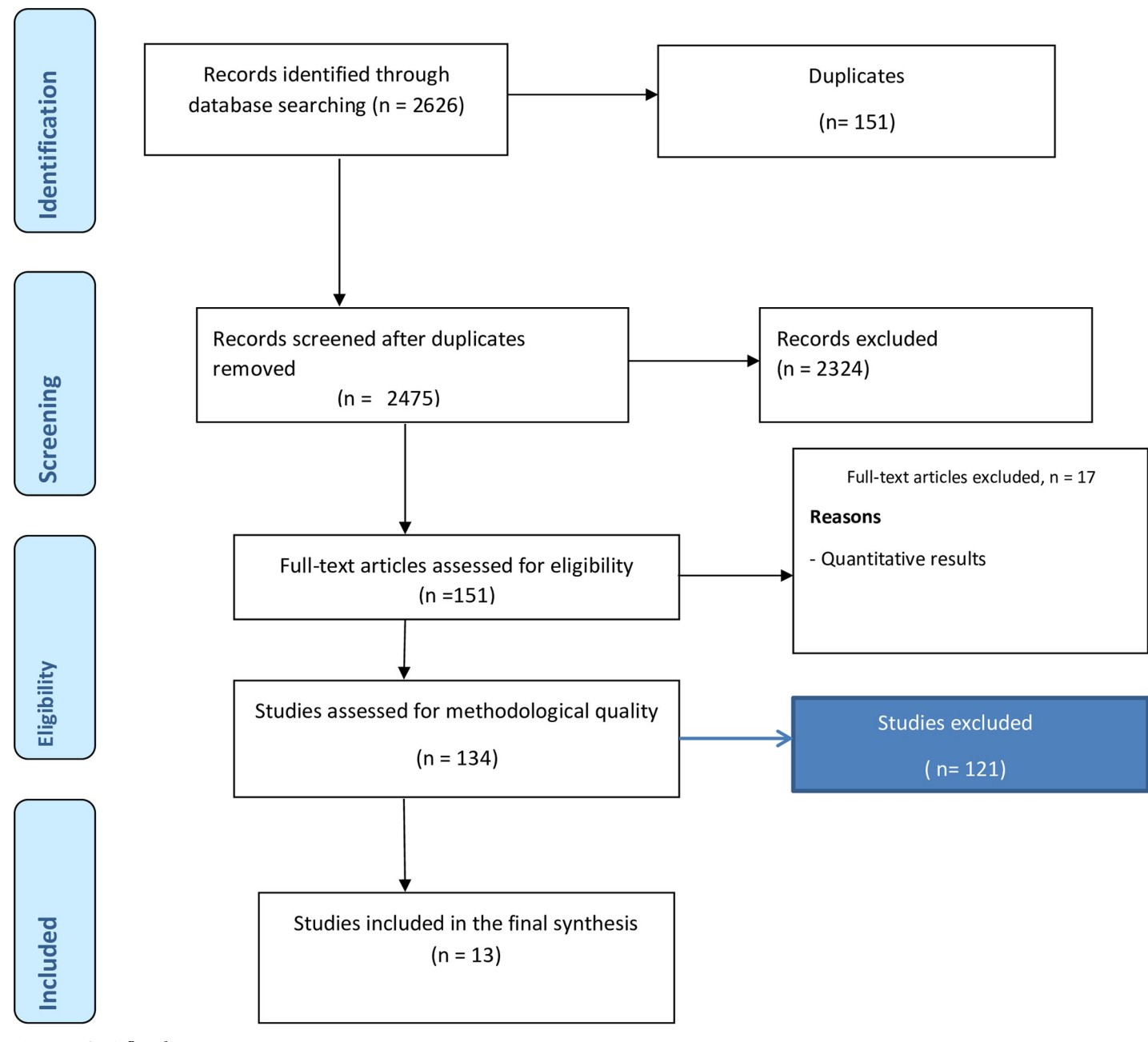

**Fig 1. PRISMA flow diagram.**

Concerning social support, three sub-themes were realised. These were: support from friends and peers, support from family, and support from other people who are neither friends nor family. Two studies reported on friends being major motivators of their contraceptive use [23, 29]. Burke et al. [29] and Hall et al. [30] also reported on the family being an important motivation for young people's contraceptive use. Tabane and Peu [23] also indicated that apart from friends, there were "significant others" who motivate young people to utilise contraceptives.

Two sub-themes were realised from the theme on the protection of identity. These were anonymity and confidentiality. These two issues were made one theme because they both deal

**Table 1. Background of studies included in the final review.**

| Author(s) | Year | Country | Contraceptives of focus | Age groups | Data collection methods | Sample size |
|---|---|---|---|---|---|---|
| Patient et al. [19] | 2000 | South Africa | Condoms | 15–24 | FGD | 88 |
| Otoide et al. [20] | 2001 | Nigeria | All | 15–24 | FDG | 149 |
| Castle [21] | 2003 | Mali | All | 15–24 | IDI | 84 |
| Flaherty et al. [22] | 2005 | Uganda | All | 14–20 | FGD and general Discussions | 29 in FGD and >500 in general discussions |
| Tabane&Peu [23] | 2015 | South Africa | All | 15–19 | IDI | 15 |
| Ochako et al. [24] | 2015 | Kenya | Modern Contraceptives | 15–24 | IDI | 34 |
| Hall et al. [25] | 2015 | Ghana | All | 15–24 | IDI | 63 |
| Hokororo et al. [26] | 2015 | Tanzania | All | 15–20 | FGD | 49 |
| Capurchande et al. [27] | 2016 | Mozambique | All | 15–24 | IDI, FGD, informal conversation, and observation | 42 |
| Hall et al. [28] | 2016 | Ghana | All | 15–24 | IDI | 63 |
| Burke et al. [29] | 2017 | Senegal | All | 18–24 (with disabilities) | IDI and FGD | 144 |
| Hall et al. [30] | 2018 | Ghana | All | 15–24 | IDI | 63 |
| Rokicki&Merten [31] | 2018 | Ghana | Contraceptive pills | 18–24 | IDI | 32 |

with concealing the identity of the young people regarding their utilization of contraceptives. Two studies made findings on anonymity [19, 29] where the study participants, for instance, indicated that the use of purchase points which were mainly vending machines offered

**Table 2. Motivations for contraceptive use among young people in SSA.**

| Main theme | Sub-theme | Studies |
|---|---|---|
| Social support | From friends and peers | Tabane & Peu [23] |
| | | Burke et al. [29] |
| | From family | Hall et al. [30] |
| | | Burke et al. [29] |
| | Others | Tabane & Peu [23] |
| Protection of identity | Anonymity (of purchase points[vending machines]) | Patient & Orr [19] |
| | | Burke et al. [29] |
| | Confidentiality (associated with patent medicine stores) | Otoide et al. [20] |
| | | Burke et al. [29] |
| Ready availability, affordability, and access | Accessibility | Patient & Orr [19] |
| | | Burke et al. [29] |
| | Availability (of condoms) | Castle [21] |
| | Affordability (low cost of condoms) | Castle [21] |
| Effectiveness in preventing unintended pregnancy and STIs | Preventing unintended pregnancy | Castle [21] |
| | | Tabane & Peu [23] |
| | | Rokiciki & Merten (2018) |
| | Preventing STIs | Castle [21] |
| Others | The need to maximise fertility in future (to gain status through child bearing) | Castle [21] |
| | Service providers being of the same sex with the young people | Flaherty et al. [22] |
| | Perceived quality of Pharmacy shops | Burke et al. [29] |

acceptable levels of anonymity and thus, motivated them to purchase and use the contraceptives. Confidentiality was also associated with patent medicine stores in 2 studies [20, 29]. Lack of confidence in sellers of modern contraceptives was a major hindrance for young people's utilization of contraceptives [20, 29].

Three studies also mentioned ready availability, affordability, and access to contraceptives as a major motivation for their utilization of contraceptives. Accessibility was for instance found by Patient and Orr [1] and Burke et al. [29] while availability and affordability were both found in Castle's [21] study. According to the young people in these studies, condoms for instance were cheap and this made it possible to afford and use them. The condoms were also readily available at various points of sale for easy purchase whenever needed.

The effectiveness of contraceptives in preventing unintended pregnancy and STIs was also reported as a motivation for their utilisation. Castle [21], Tabane and Peu [23], and Rokicki and Merten [31] all made findings on the effectiveness of contraceptives in preventing unintended pregnancy. Castle [21] also reported that a major motivation for using condoms among young people was the fact that it made it possible for them to avoid contracting STIs. Other motivations mentioned by the studies comprised the need to maximise fertility in future (to gain status through childbearing) [21], service providers being of the same sex as the young people [22], and perceived quality of pharmacy shops [29].

## Barriers to contraceptive use

Table 3 presents the barriers to contraceptive use which were reported by all the 13 studies included in our analyses. Three main themes were identified from the review. These were personal, societal, and health systems-based barriers. With personal barriers, five sub-themes were observed. They were: myths and misconceptions, lack/inadequate knowledge, negative attitude towards contraceptive use, known side effects of contraceptives, and financial challenges. Myths and misconceptions were the most reported personal barriers to contraceptive use. The sub-themes reported under myths and perceptions were lack of trust in contraceptives (especially condoms), colour and size of available condoms considered unsuitable, contraceptive use encourages promiscuity and straying (cheating), contraceptive use reduces sexual pleasure, risk of future infertility with contraceptive use, and perceived ineffectiveness of contraceptives in preventing conception. Five studies also reported the known side effects of contraceptives including weight gain, bleeding, high blood pressure, headache, and disruption of the menstrual cycle as personal barriers to contraceptive use among the young people.

Societal-based barriers were reported by nine studies, comprising two sub-themes which were social consequences of contraceptive use and social norms. The specific societal based barriers to contraceptive use were: divorce, accusations of witchcraft, stigma, the tag of being promiscuous, disproval of contraceptive use by friends and colleagues as well as family and the larger society, societal prohibition of discussions on issues concerning contraception, contraception being considered an issue only for females, and religious prohibitions.

Health systems-based barriers which were reported by 7 of the studies comprised five sub-themes; lack of privacy and confidentiality at health facilities, negative attitude of health professionals, long waiting time, poor communication between health professionals and young people, and physical inaccessibility of buildings by the persons with disability as the sub-themes. The most-reported sub-theme was the negative attitude of health professionals which comprised being treated disrespectfully, being entirely refused contraceptive services, being denied teaching about contraceptives, and discrimination.

**Table 3. Barriers to contraceptive use among young people in SSA.**

| Main theme | Sub-theme | Codes | Studies |
|---|---|---|---|
| Personal | Myths and misconceptions | Lack of trust in contraceptives (especially condoms) | Patient & Orr [19] |
| | | | Rokicki & Merten [31] |
| | | Colour and size of available condoms considered unsuitable | Patient & Orr [19] |
| | | Contraceptive use encourages promiscuity and straying (cheating) | Ochako et al. [24] |
| | | | Burke et al. [29] |
| | | | Rokicki & Merten [31] |
| | | Contraceptive use reduces sexual pleasure | Ochako et al. [24] |
| | | | Capurchande et al. [27] |
| | | | Rokicki & Merten [31] |
| | | There is a risk of future infertility with contraceptive use | Otoide et al. [20] |
| | | | Castle [21] |
| | | | Flaherty et al. [22] |
| | | | Ochako et al. [24] |
| | | Perceived ineffectiveness of contraceptives in preventing conception | Otoide et al. [20] |
| | | | Hokororo et al. [26] |
| | | | Capurchande et al. [27] |
| | | | Rokicki & Merten [31] |
| | Lack/Inadequate knowledge | Poor knowledge on the mechanism of action of contraceptives and on how to utilise them | Otoide et al. [20] |
| | | | Flaherty et al. [22] |
| | | | Tabane & Peu [23] |
| | | | Hokororo et al. [26] |
| | | | Capurchande et al. [27] |
| | | | Rokicki & Merten [31] |
| | | Lack of reliable, trusted and non-judgemental sources of information on contraceptives | Flaherty et al. [22] |
| | | | Burke et al. [29] |
| | | | Rokicki & Merten [31] |
| | Negative attitude towards contraceptive use | Lack of personal motivation and willingness to utilise contraception | Tabane & Peu [23] |
| | | Contraceptive use (mainly condom) is boring, stressful, too much of a responsibility, and clinical | Patient & Orr [19] |
| | | | Tabane & Peu [23] |
| | | | Capurchande et al. [27] |
| | Known side effects of contraceptives | Weight gain, headache, bleeding, high blood pressure, and disruption of the menstrual cycle | Otoide et al. [20] |
| | | | Tabane & Peu [23] |
| | | | Ochako et al. [24] |
| | | | Capurchande et al. [27] |
| | | | Rokicki & Merten [31] |
| | Financial challenges | Unaffordability of contraceptives and contraceptive services | Burke et al. [29] |
| | | | Rokicki & Merten [31] |

(*Continued*)

**Table 3.** (Continued)

| Main theme | Sub-theme | Codes | Studies |
|---|---|---|---|
| Societal based | Social consequences of contraceptive use | Divorce | Castle [21] |
| | | Accusations of witchcraft | Castle [21] |
| | | Stigma | Hall et al. [25] |
| | | | Hokororo et al. [26] |
| | | | Capurchande et al. [27] |
| | | | Hall et al. [28] |
| | | The tag of being promiscuous | Hall et al. [30] |
| | Social norms | Disproval of contraceptive use by friends and colleagues | Tabane & Peu [23] |
| | | Disproval of contraceptive use by family and the larger society | Tabane & Peu [23] |
| | | | Burke et al. [29] |
| | | | Hall et al. [30] |
| | | Societal prohibition of discussions on issues concerning contraception | Capurchande et al. [27] |
| | | | Burke et al. [29] |
| | | | Rokicki & Merten [31] |
| | | Contraception being considered an issue only for females | Capurchande et al. [27] |
| | | Religious prohibitions | Burke et al. [29] |
| Health systems-based | Lack of privacy and confidentiality at health facilities | | Flaherty et al. [22] |
| | | | Hokororo et al. [26] |
| | | | Burke et al. [29] |
| | Negative attitude of health professionals | Being treated disrespectfully | Flaherty et al. [22] |
| | | Being entirely refused contraceptive services | Flaherty et al. [22] |
| | | | Hokororo et al. [26] |
| | | Being denied teaching about contraceptives | Tabane & Peu [23] |
| | | Discrimination | Burke et al. [29] |
| | | | Hall et al. [30] |
| | Long waiting time | | Hokororo et al. [26] |
| | Poor communication between health professionals and young people | Overly technical language used at health facilities | Capurchande et al. [27] |
| | | | Burke et al. [29] |
| | | Power asymmetry in communication between health professionals and young people | Capurchande et al. [27] |
| | Physical inaccessibility of health facilities | Staircases are unfriendly having to be accompanied by someone to facilitate access | Burke et al. [29] |

## Discussion

In this review, we explored the barriers and motivators of contraceptive use among younger people in SSA using 13 qualitative publications. The motivators were re-categorised into social support, protection of identity, ready availability, affordability and access, effectiveness in preventing unintended pregnancy and STIs, and other motivators. The barriers were also re-categorised into personal, societal, and health systems-based barriers.

Social support was realised in our review as an important motivator of contraceptive use. Tabane and Peu [23] for instance posited that approval from friends and positive peer influence from peers greatly encourage young people to utilise contraceptives. Burke et al. [29] and Hall et al. [30] also noted that family support is an instrumental protective factor for young

people's utilisation of contraceptives. We also realised from our review that when anonymity and confidentially are assured and there is ready availability, affordability, and access, young people are highly motivated to patronise contraceptives and contraceptive services. This is an issue of importance because, young people in SSA have been known to have grave unmet needs for contraceptive use which stems from their inability to access, afford, and/or be guaranteed confidentiality when they make efforts to utilise contraceptives [32, 33].

Despite the motivators realised in our review, we observed that myriad of barriers also inhibit young people from utilising contraceptives and this has negative implications for SSA countries towards the attainment of health for all by the year 2030. Key among these challenges are myths and misconceptions. Young people according to Otoide et al. [20], Hokororo et al. [26], Capurchande et al. [27], and Rokicki and Merten [31] for instance, erroneously perceived that contraceptives were ineffective in preventing conception. They also believed that contraceptive use posed risks of future infertility [20–22, 24]. The myth and misconceptions realised in our review could have been due to lack of knowledge which Otoide et al. [20], Flaherty et al. [22], Tabane and Peu [23], Hokororo et al. [26], Capurchande et al. [27], Burke et al. [29] and Rokicki and Merten [31] found as comprising poor knowledge on the mechanism of action of contraceptives and on how to utilise them and lack of reliable, trusted and non-judgmental sources of information on contraceptives was also reported by seven of the studies. The findings of our review regarding knowledge point to its essential role in the reproductive health decision making of young people including contraception [34–37].

Our review showed that the known side effects of contraception such as weight gain, bleeding, headaches, high blood pressure, and disruption of the menstrual cycle were personal barriers to contraceptive use among the young people. If such fears are not effectively managed through innovative approaches which reassure them of the important reasons for utilisation, such young people would continue to shy away from contraceptive use and this would not auger well for the achievement of universal health coverage by countries in SSA. Besides, financial challenges which were mainly about the unaffordability of contraceptives and contraceptive services remained major barriers to the young people in their quest to utilise the contraceptives according to Burke et al. [29] and Rokicki and Merten [31]. This finding, however, contradicts arguments by Castle [21] that young people in her study felt contraceptives were affordable and this served as a motivation for them in utilizing such contraceptives.

Attitude of health professionals towards clients is an important determinant of health care utilisation [38–40]. The negative attitude of health professionals towards young people was reported by Flaherty et al. [22], Tabane and Peu [23], Hokororo et al. [26], Burke et al. [29], and Hall et al. [30] as a health systems-based barrier to contraceptive use among young people and this manifested in being treated disrespectfully, being entirely refused contraceptive services, being denied teaching about contraceptives, and discrimination. Our finding of physical inaccessibility of buildings as reported by Burke et al. [29] where staircases were unfriendly and the physically challenged young people needing to be accompanied by someone, points to the general neglect of the rights of persons with physical disabilities to these buildings [41]. Our finding thus justifies calls for the physical environment to be designed and equipped to meet the needs of persons with disabilities (PWDs) and prioritization of their needs through national planning, budgeting and other national programmes.

Social norms are powerful societal factors which predict the health-seeking behaviour of people. In our review, we realised that these norms militated against the utilisation of contraceptives among young people in the form of disproval of contraceptive use by friends and colleagues, disproval of contraceptive use by family and the larger society, societal prohibition of discussions on issues concerning contraception, and contraception being considered an issue only for females. The continuous prevalence of such inhibitive social norms in SSA would only

continue to preclude young people in these countries from utilizing contraceptives and thus making it impossible for the countries to achieve universal levels of contraceptive use.

In most SSA countries, major contraceptive interventions have included the promotion and delivery of contraceptives during the postpartum period, expansion of long-acting contraceptive options, and community-based contraceptive outreach and service delivery [42, 43]. The effectiveness of these interventions has, however, been a major challenge as they mainly do not yield the expected results for which they are implemented which warrant the adoption of alternative interventions proven to be effective [43, 44].

## Conclusions

Our review revealed social support, protection of identity, ready availability, affordability and access, effectiveness in preventing unintended pregnancy and STIs as motivators of young people's utilization of contraceptives in SSA. Despite the availability of these motivators, myriad of personal, societal, and health systems-based barriers prevail and prevent the young people from utilizing the contraceptives. These barriers include myths and misconceptions, lack/inadequate knowledge on contraceptives, known side effects of contraceptives, financial challenges, prohibitive social norms, negative attitude of health professionals, physical inaccessibility of buildings, and poor communication between health professionals and young people. The perpetuation of these barriers implies that many SSA countries, especially those from which the barriers where identified may not be able to achieve the SDG target of health for all by the year 2030.

## Recommendations

To ameliorate the barriers to contraceptive use and accelerate the progress of SSA countries towards the achievement of health for all by the year 2030, we proffer these policy recommendations:

1. There is a need for community engagements to improve prohibitive social norms to make them more receptive towards contraceptive use and discussions regarding contraception.

2. Contraceptive service provision should be made friendly in countries which are not already implementing such young people-friendly contraceptive services. In such countries, there should, for instance, be a re-conscientization of health professionals to improve their attitudes towards young people in the provision of contraceptive services to them. Privacy and confidentiality should be afforded the young people in their utilisation of services.

3. Efforts should be made by respective SSA countries to make public health facilities accessible for persons living with disabilities.

4. To reduce financial barriers, voucher schemes which have been implemented in some countries, could be adopted by other countries for young people to access subsidised or free contraceptive services.

## Strengths and limitations

Our study is the first effort at exploring the barriers and motivations for contraception among young people using purely qualitative studies. The fact that we reviewed only qualitative studies, however, meant that other relevant papers reporting barriers and motivations for contraceptive use among young people were expunged. It is, however, noteworthy that such studies could have contributed to a better understanding of the totality of the barriers and motivators for young people's contraceptive use.

## Supporting information

**S1 File. PRISMA checklist.**
(DOC)

**S2 File. List of articles included in the review.**
(DOCX)

## Author Contributions

**Conceptualization:** Luchuo Engelbert Bain, Hubert Amu, Elvis Enowbeyang Tarkang.

**Data curation:** Luchuo Engelbert Bain, Hubert Amu, Elvis Enowbeyang Tarkang.

**Formal analysis:** Luchuo Engelbert Bain, Hubert Amu.

**Methodology:** Luchuo Engelbert Bain, Hubert Amu, Elvis Enowbeyang Tarkang.

**Validation:** Hubert Amu, Elvis Enowbeyang Tarkang.

**Writing – original draft:** Luchuo Engelbert Bain, Hubert Amu, Elvis Enowbeyang Tarkang.

**Writing – review & editing:** Luchuo Engelbert Bain, Hubert Amu, Elvis Enowbeyang Tarkang.

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
