## [Decision Letter · Decision Letter 0]

20 Oct 2020

PONE-D-20-10773

Barriers to and motivations for contraceptive use among adolescents and young women in Sub-Saharan Africa: A systematic review of qualitative studies

PLOS ONE

Dear Dr. Luchuo Engelbert Bain,

Thank you for submitting your manuscript to PLOS ONE. After careful consideration, we feel that it has merit but does not fully meet PLOS ONE’s publication criteria as it currently stands. Therefore, we invite you to submit a revised version of the manuscript that addresses the points raised during the review process.

Kindly revise your manuscript taking into cognisance the comments of the reviewers.

We look forward to receiving your revised manuscript.

Kind regards,

Eugene Kofuor Maafo Darteh, Ph.D.

Academic Editor

PLOS ONE

Journal Requirements:

2. Please address the following:

- Please include some discussion of the quality/biases of each study.

- Please include the full electronic search strategy for at least one database as supplementary information.

3.Thank you for stating the following financial disclosure:

 [NO : The funders had no role in study design, data collection and analysis, decision to publish, or preparation of the manuscript].

4.We note that you have indicated that data from this study are available upon request. PLOS only allows data to be available upon request if there are legal or ethical restrictions on sharing data publicly. For information on unacceptable data access restrictions, please see http://journals.plos.org/plosone/s/data-availability#loc-unacceptable-data-access-restrictions.

Reviewers' comments:

Reviewer's Responses to Questions

**Comments to the Author**

1. Is the manuscript technically sound, and do the data support the conclusions?

Reviewer #1: Yes

Reviewer #2: Yes

2. Has the statistical analysis been performed appropriately and rigorously? 

Reviewer #1: N/A

Reviewer #2: N/A

3. Have the authors made all data underlying the findings in their manuscript fully available?

Reviewer #1: Yes

Reviewer #2: Yes

4. Is the manuscript presented in an intelligible fashion and written in standard English?

Reviewer #1: Yes

Reviewer #2: No

5. Review Comments to the Author

Reviewer #1: Authors need to go through the PDF document of the manuscript where reviewer comments are included.

The objective/question of the review was on three areas but only two of the questions were answered with the review. One question remains to be answered.

Reviewer #2: This is a systematic review on Barriers to and motivations for contraceptive use among adolescents and young women in Sub-Saharan Africa. Since low contraceptive usage is a challenge African countries are grabbling with, synthesising knowledge to identify gaps is important for public health.

Introduction: Pg3 The age range for young people includes adolescents. So why separate adolescents and young people in your population of interest?

Methods: Pg 4 "the reviewers met" does not connote an objective process. Why was it not done independently and if necessary a third person adjudicates in the event of indecision?

Policy recommendations: There is no indication from your write up on what policies countries have in place with regards to contraceptive usage for readers to appreciate if your policy recommendations are necessary. As it is no one knows what the countries within the review are doing.

General comments:

1. There are so many recommendations authors could give. For instance taking a look at Table 1, the age groups of included studies indicate a gap. As indicated in the introduction the age group is 10-24 years but the selected studies lack on age group 10-13 who are also sexually active.

2. One would have expected that as what is known is being synthesised authors will identify gaps that need closing with research to win the low contraceptive battle. Because, bringing together what is available in the articles is not enough. This is concluded like a research based on a primary data. Gaps must be identified. Therefore as it is, a major revision is needed.

3. There's the need for editing to correct typographical errors

4. Authors failed to indicate why the focus is on qualitative studies

6. PLOS authors have the option to publish the peer review history of their article (what does this mean?). If published, this will include your full peer review and any attached files.

Reviewer #1: **Yes: **Hailemariam Segni A. (MD, MPH, Gyn/Obs)

Reviewer #2: No

---

## [Author Response · Author response to Decision Letter 0]

2 Dec 2020

Response to Reviewers

a. Please clarify the sources of funding (financial or material support) for your study. List the grants or organizations that supported your study, including funding received from your institution.

Response: This study did not receive any funding.

Response: No funding was received for this study

Response: No salaries were received from funders. 

d. If you did not receive any funding for this study, please state: “The authors received no specific funding for this work.”

Response: “The authors received no specific funding for this work.”

Response:

4.We note that you have indicated that data from this study are available upon request. PLOS only allows data to be available upon request if there are legal or ethical restrictions on sharing data publicly. For information on unacceptable data access restrictions, please see http://journals.plos.org/plosone/s/data-availability#loc-unacceptable-data-access-restrictions.

 Response:

Response:

5. Review Comments to the Author

Reviewer #1: 

Authors need to go through the PDF document of the manuscript where reviewer comments are included.

Response: this has been done and all issues raised by the reviewer have been pasted in the “response to reviewers” document and duly addressed

The objective/question of the review was on three areas but only two of the questions were answered with the review. One question remains to be answered.

1. Search was conducted from a small number of data bases were searched.

Response: Thank you for this comment. Based on previous studies including those published in Plos One, we believe that the number of databases searched were exhaustive enough to ensure we retrieve all relevant publications. We actually searched PubMed, EMBASE, Ebsco/PsycINFO and Scopus. Searching other sources may only create further duplication of papers as these are the most popular databases available.

2. Too many papers (4457) were retrieved from which only 13 were eligible. Maybe the search terms used were not refined enough?

Response: Our study focused only on qualitative studies which was the reason for the rather small number of papers included in the final review. We have also detailed the search tersm in the methods section and believe that the terms used were exhaustive enough to produce the expected results.

3. The review has not answered one of the questions, Prevalence and usage.

Response: The focus of this systematic review was only on the barriers and motivations for contraceptive uptake among young people.

4. The protocol mentioned Prevalence and usage as one of the questions to be answered.

Response: The main review has focused only on the barriers and motivations

5. Enough number of data bases were not searched from.

Response: Thank you for this comment. Based on previous studies including those published in Plos One, we believe that the number of databases searched were exhaustive enough to ensure we retrieve all relevant publications. We actually searched PubMed, EMBASE, Ebsco/PsycINFO and Scopus. Searching other sources may only create further duplication of papers as these are the most popular databases available.

6. This maybe one of the reasons why you had too many non-eligible studies during the search. Restriction of time of study and/or publication is one thing to seriously consider.

Response: We agree with the reviewer that this is a plausible explanation of the high number of non – eligible studies we had. However, empirical qualitative research on the subject was not very rampant, and we wanted to be an inclusive as possible.

7. What about during a situation of failure to reach at consensus between the two reviewers? You need to have a third tie breaker reviewer in the team.

Response: Even though a third reviewer was on standby, no such disagreements emerged between the two reviewers which required his inputs. That is why it has not been mentioned in the manuscript.

8. Prevalence and usage?

Response: This has not been explored in our current review even though contained in the registered protocol.

9. Physical inaccessibility of buildings by the persons with disability" should be discussed here as one of the major barriers.

Response: This has been done. See page 11.

10. "Physical inaccessibility of buildings by the persons with disability" should be part of the conclusion section as a major barrier.

Response: This has been done. See page 12.

11. It is very difficult to reach at this type of conclusion based on this systematic review.

Response: We have refined the statement to indicate that some countries, especially the ones from which the studies were reviewed, may not be able to achieve the SDG target with the persistence of the barriers identified. See page 12.

Reviewer #2: 

This is a systematic review on Barriers to and motivations for contraceptive use among adolescents and young women in Sub-Saharan Africa. Since low contraceptive usage is a challenge African countries are grabbling with, synthesising knowledge to identify gaps is important for public health.

Introduction: Pg3 The age range for young people includes adolescents. So why separate adolescents and young people in your population of interest?

Response: We have revised the manuscript to focus only on young people.

Methods: Pg 4 "the reviewers met" does not connote an objective process. Why was it not done independently and if necessary a third person adjudicates in the event of indecision?

Response: the phrased has been revised accordingly (See page 5). Also, while a third reviewer was actually on standby, no serious disagreements in terms of what to include in the final review emerged and so that reviewer did not intervene.

Policy recommendations: There is no indication from your write up on what policies countries have in place with regards to contraceptive usage for readers to appreciate if your policy recommendations are necessary. As it is no one knows what the countries within the review are doing.

Response: A writ-up has now been included in the discussion in this regard. See page 12.

General comments:

1. There are so many recommendations authors could give. For instance taking a look at Table 1, the age groups of included studies indicate a gap. As indicated in the introduction the age group is 10-24 years but the selected studies lack on age group 10-13 who are also sexually active.

Response: A recommendation has been provided in this regard. See page 14.

2. One would have expected that as what is known is being synthesised authors will identify gaps that need closing with research to win the low contraceptive battle. Because, bringing together what is available in the articles is not enough. This is concluded like a research based on a primary data. Gaps must be identified. Therefore as it is, a major revision is needed.

3. There's the need for editing to correct typographical errors

Response: This has been done. An English language expert has proof-read the manuscript.

4. Authors failed to indicate why the focus is on qualitative studies.

Response: This has been done. See page 5.

---

## [Decision Letter · Decision Letter 1]

12 Jan 2021

PONE-D-20-10773R1

Barriers to and motivations for contraceptive use among young people in Sub-Saharan Africa: A systematic review of qualitative studies

PLOS ONE

Dear Dr. Luchuo Engelbert Bain,

Thank you for submitting your manuscript to PLOS ONE. After careful consideration, we feel that it has merit but does not fully meet PLOS ONE’s publication criteria as it currently stands. Therefore, we invite you to submit a revised version of the manuscript that addresses the points raised during the review process.

Kindly address the minor comments of the reviewers.

Please submit your revised manuscript by 11th March 2021 If you will need more time than this to complete your revisions, please reply to this message or contact the journal office at plosone@plos.org. Please include the following items when submitting your revised manuscript:

We look forward to receiving your revised manuscript.

Kind regards,

Eugene Kofuor Maafo Darteh, Ph.D.

Academic Editor

PLOS ONE

Reviewers' comments:

Reviewer's Responses to Questions

**Comments to the Author**

1. If the authors have adequately addressed your comments raised in a previous round of review and you feel that this manuscript is now acceptable for publication, you may indicate that here to bypass the “Comments to the Author” section, enter your conflict of interest statement in the “Confidential to Editor” section, and submit your "Accept" recommendation.

Reviewer #1: All comments have been addressed

Reviewer #2: (No Response)

2. Is the manuscript technically sound, and do the data support the conclusions?

Reviewer #1: Yes

Reviewer #2: Yes

3. Has the statistical analysis been performed appropriately and rigorously? 

Reviewer #1: Yes

Reviewer #2: N/A

4. Have the authors made all data underlying the findings in their manuscript fully available?

Reviewer #1: Yes

Reviewer #2: Yes

5. Is the manuscript presented in an intelligible fashion and written in standard English?

Reviewer #1: Yes

Reviewer #2: (No Response)

6. Review Comments to the Author

Reviewer #1: Reviewer’s comments 2

• Title: it is better if written as “Barriers and motivators of contraceptive use among young people in Sub Saharan Africa: a systematic review of qualitative studies”.

• Key words: replace “Motivation” with “Motivators”.

• Methods: you need to include in methods section that a third reviewer was standby to be involved if disagreement happens between the two reviewers but there was need for it.

• Some editorial work is still needed.

Reviewer #2: Please I do not see the response authors indicated to be on page 12 of the manuscript to be a response to the request to provide existing policies on the various countries from where studies have been included. This is to contextualise and make relevant the policy recommendations.

7. PLOS authors have the option to publish the peer review history of their article (what does this mean?). If published, this will include your full peer review and any attached files.

Reviewer #1: **Yes: **Hailemariam Segni Abawollo, MD/OB-GYN, MPH

Reviewer #2: No

---

## [Author Response · Author response to Decision Letter 1]

17 Jan 2021

RESPONSE TO REVIEWERS 

Reviewer #1: Reviewer’s comments 2

• Title: it is better if written as “Barriers and motivators of contraceptive use among young people in Sub Saharan Africa: a systematic review of qualitative studies”.

Response: The title has been revised as recommended by the reviewer. See page 1

• Key words: replace “Motivation” with “Motivators”.

Response: The change has been made as the reviewer recommended. See page 3

• Methods: you need to include in methods section that a third reviewer was standby to be involved if disagreement happens between the two reviewers but there was need for it.

Response: The addition has been made. See page 6

• Some editorial work is still needed.

Response: Some more editing has been done on the manuscript

Reviewer #2: Please I do not see the response authors indicated to be on page 12 of the manuscript to be a response to the request to provide existing policies on the various countries from where studies have been included. This is to contextualise and make relevant the policy recommendations.

Response: Please see page 13, lines 271-276

---

## [Decision Letter · Decision Letter 2]

11 Apr 2021

PONE-D-20-10773R2

Barriers and motivators of contraceptive use among young people in Sub Saharan Africa: a systematic review of qualitative studies

PLOS ONE

Dear Dr. Luchuo Engelbert Bain,

Thank you for submitting your manuscript to PLOS ONE. After careful consideration, we feel that it has merit but does not fully meet PLOS ONE’s publication criteria as it currently stands. Therefore, we invite you to submit a revised version of the manuscript that addresses the points raised during the review process.

You are requested to revise the manuscript focusing on the recommendations. Ensure that the recommendations are reason and doable.  

We look forward to receiving your revised manuscript.

Kind regards,

Eugene Kofuor Maafo Darteh, Ph.D.

Academic Editor

PLOS ONE

Journal Requirements:

Reviewers' comments:

Reviewer's Responses to Questions

**Comments to the Author**

1. If the authors have adequately addressed your comments raised in a previous round of review and you feel that this manuscript is now acceptable for publication, you may indicate that here to bypass the “Comments to the Author” section, enter your conflict of interest statement in the “Confidential to Editor” section, and submit your "Accept" recommendation.

Reviewer #2: (No Response)

2. Is the manuscript technically sound, and do the data support the conclusions?

Reviewer #2: (No Response)

3. Has the statistical analysis been performed appropriately and rigorously? 

Reviewer #2: N/A

4. Have the authors made all data underlying the findings in their manuscript fully available?

Reviewer #2: Yes

5. Is the manuscript presented in an intelligible fashion and written in standard English?

Reviewer #2: No

6. Review Comments to the Author

Reviewer #2: Materials and methods

1. Please correct the sentence in line #101-103

Recommendations

1. Line #275: There is an omission in the sentence

2. Recommendation 1(line #273-276): but literature showed knowledge does not always translate to action, so authors should be recommending something other than increasing knowledge

3. Recommendation 3 (line #279-282): I know a country included in the review that is already practicing this, so you can’t box them together to still do this. Unless the country is facing implementation challenges. Then you suggest a separate recommendation to that effect.

4. Recommendation 5 (line#285-286): a country in your review has begun implementing free FP in pockets of places and not scaled up nationwide though. But your suggestion implies all countries should implement without considering what others are doing.

5. It is for these reasons I suggested you provide a context on the countries (i.e. policy) from which studies are included for your audience to appreciate things better.

7. PLOS authors have the option to publish the peer review history of their article (what does this mean?). If published, this will include your full peer review and any attached files.

Reviewer #2: No

---

## [Author Response · Author response to Decision Letter 2]

19 Apr 2021

6. Review Comments to the Author

Reviewer #2: Materials and methods

1. Please correct the sentence in line #101-103

Response: The correction has been made as the reviewer suggested (See page 6) 

Recommendations

1. Line #275: There is an omission in the sentence

Response: The entire sentence has been deleted in response to a subsequent comment made by the reviewer. See page 14.

2. Recommendation 1(line #273-276): but literature showed knowledge does not always translate to action, so authors should be recommending something other than increasing knowledge

Response: The recommendation has been deleted based on the reviewer’s comments.

3. Recommendation 3 (line #279-282): I know a country included in the review that is already practicing this, so you can’t box them together to still do this. Unless the country is facing implementation challenges. Then you suggest a separate recommendation to that effect.

Response: We have reviewed the suggestion to focus on countries which are not already implementing friendly contraceptive services for young people. See page 15.

4. Recommendation 5 (line#285-286): a country in your review has begun implementing free FP in pockets of places and not scaled up nationwide though. But your suggestion implies all countries should implement without considering what others are doing.

Response: We have revised the recommendation to focus on countries which are not already implementing the voucher schemes.

5. It is for these reasons I suggested you provide a context on the countries (i.e. policy) from which studies are included for your audience to appreciate things better

Response: We have deleted the recommendation. See page 15.

---

## [Decision Letter · Decision Letter 3]

24 May 2021

Barriers and motivators of contraceptive use among young people in Sub Saharan Africa: a systematic review of qualitative studies

PONE-D-20-10773R3

Dear Dr. Luchuo Engelbert Bain,

We’re pleased to inform you that your manuscript has been judged scientifically suitable for publication and will be formally accepted for publication once it meets all outstanding technical requirements.

Kind regards,

Eugene Kofuor Maafo Darteh, Ph.D.

Academic Editor

PLOS ONE

Additional Editor Comments (optional):

Reviewers' comments:

Reviewer's Responses to Questions

**Comments to the Author**

1. If the authors have adequately addressed your comments raised in a previous round of review and you feel that this manuscript is now acceptable for publication, you may indicate that here to bypass the “Comments to the Author” section, enter your conflict of interest statement in the “Confidential to Editor” section, and submit your "Accept" recommendation.

Reviewer #2: All comments have been addressed

2. Is the manuscript technically sound, and do the data support the conclusions?

Reviewer #2: Yes

3. Has the statistical analysis been performed appropriately and rigorously? 

Reviewer #2: Yes

4. Have the authors made all data underlying the findings in their manuscript fully available?

Reviewer #2: Yes

5. Is the manuscript presented in an intelligible fashion and written in standard English?

Reviewer #2: Yes

6. Review Comments to the Author

Reviewer #2: (No Response)

7. PLOS authors have the option to publish the peer review history of their article (what does this mean?). If published, this will include your full peer review and any attached files.

Reviewer #2: No

---

## [Editor Report · Acceptance letter]

27 May 2021

PONE-D-20-10773R3 

Barriers and motivators of contraceptive use among young people in Sub-Saharan Africa: A systematic review of qualitative studies 

Dear Dr. Engelbert Bain:

I'm pleased to inform you that your manuscript has been deemed suitable for publication in PLOS ONE. Congratulations! Your manuscript is now with our production department. 

Kind regards, 

on behalf of

Dr. Eugene Kofuor Maafo Darteh 

Academic Editor

PLOS ONE